evolution, developmental biology

variation, insect, ovary, morphology, development, evolution

**Authors for correspondence:**
Samuel H. Church
e-mail: church@g.harvard.edu
Cassandra G. Extavour
e-mail: extavour@oeb.harvard.edu

# Repeated loss of variation in insect ovary morphology highlights the role of development in life-history evolution

Samuel H. Church[1], Bruno A. S. de Medeiros[1,2], Seth Donoughe[1,3], Nicole L. Márquez Reyes[4] and Cassandra G. Extavour[1,5]

[1]Department of Organismic and Evolutionary Biology, Harvard University, Cambridge, MA 02138, USA
[2]Smithsonian Tropical Research Institute, Panama City, Panama
[3]Department of Molecular Genetics and Cell Biology, University of Chicago, Chicago, IL 60637, USA
[4]Department of Biology, Universidad de Puerto Rico en Cayey, Cayey 00736, Puerto Rico
[5]Department of Molecular and Cellular Biology, Harvard University, Cambridge, MA 02138, USA

SHC, 0000-0002-8451-103X; BASdM, 0000-0003-1663-668X; SD, 0000-0002-4773-5739; CGE, 0000-0003-2922-5855

The number of offspring an organism can produce is a key component of its evolutionary fitness and life history. Here we perform a test of the hypothesized trade-off between the number and size of offspring using thousands of descriptions of the number of egg-producing compartments in the insect ovary (ovarioles), a common proxy for potential offspring number in insects. We find evidence of a negative relationship between egg size and ovariole number when accounting for adult body size. However, in contrast to prior claims, we note that this relationship is not generalizable across all insect clades, and we highlight several factors that may have contributed to this size-number trade-off being stated as a general rule in previous studies. We reconstruct the evolution of the arrangement of cells that contribute nutrients and patterning information during oogenesis (nurse cells), and show that the diversification of ovariole number and egg size have both been largely independent of their presence or position within the ovariole. Instead, we show that ovariole number evolution has been shaped by a series of transitions between variable and invariant states, with multiple independent lineages evolving to have almost no variation in ovariole number. We highlight the implications of these invariant lineages on our understanding of the specification of ovariole number during development, as well as the importance of considering developmental processes in theories of life-history evolution.

## 1. Introduction

Offspring number is a fundamental parameter in the study of life history [1]. This number differs widely between organisms [1], and its variation is the foundation for several hypotheses about life-history evolution, including the prediction that there is an evolutionary trade-off between the number of offspring and their size (e.g. egg size) [1–3]. In insects, the number of egg-producing compartments in the ovary, called ovarioles, has been used as a proxy for potential offspring number in the study of life history [4–6]. However, without an understanding of the phylogenetic distribution of ovariole number, this hypothesized relationship cannot be assessed across insects. Here, we tested for the presence of a general trade-off between ovariole number and egg size by collecting thousands of records of ovariole number from the published literature, placing them in a phylogenetic context, and comparing them to other datasets of insect reproductive morphology.

The insect female reproductive system includes a pair of ovaries, each of which contains a number of ovarioles [7] (figure 1a). Each ovariole consists of an anterior germarium containing the stem cell niche or resting oogonia,

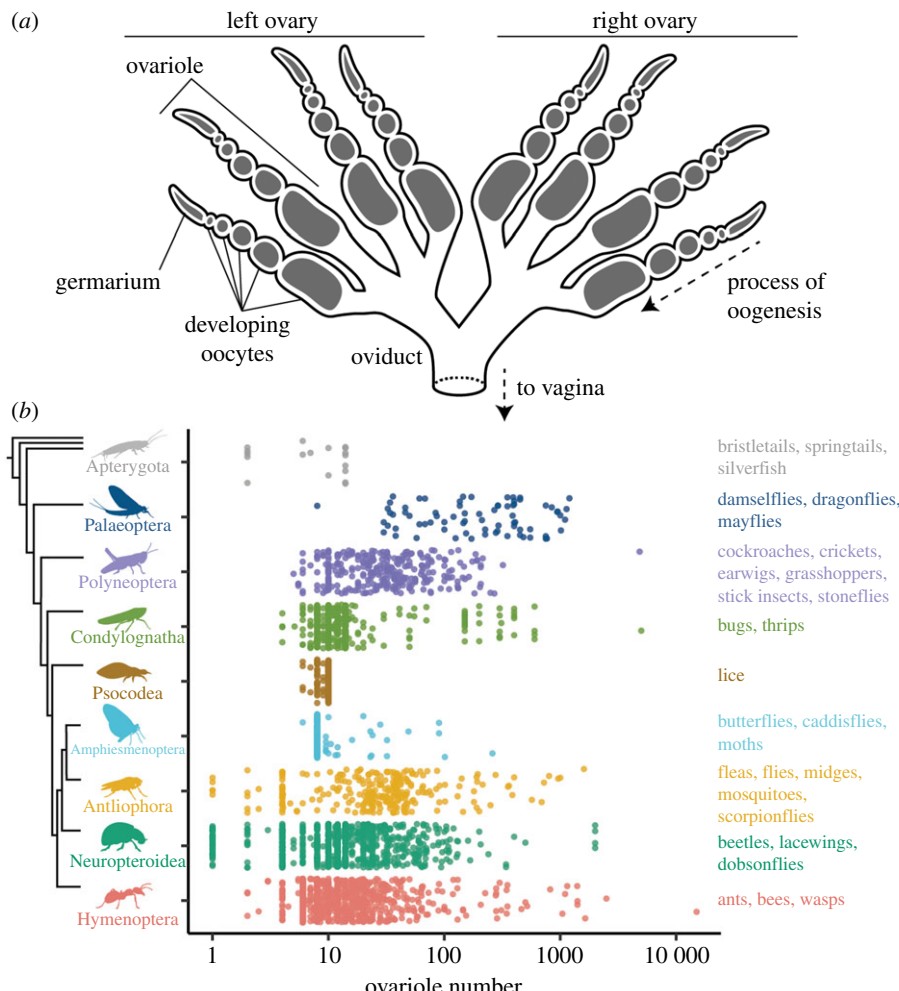

**Figure 1.** The diversity of ovariole number across insects. (*a*) Schematic of a generalized insect female reproductive system, showing a pair of ovaries, each with four ovarioles. (*b*) The range of total adult ovariole number, $\log_{10}$ scale, across nine groups of insects, arranged with random jitter on the *y*-axis within each group. Groups are, from top to bottom: Apterygota, Palaeoptera, Polyneoptera, Condylognatha, Psocodea, Amphiesmenoptera, Antliophora, Neuropteroidea and Hymenoptera.

developing oocytes arranged in an ontogenic series from anterior to posterior, and a posterior connection to a common oviduct. The number of ovarioles varies across species [6], and can vary across individuals in a population [4], as well as between the left and right ovary within a single individual [8]. Therefore, total ovariole number may be an even or odd integer for an individual female insect. In addition to variation in the number of ovarioles, the tissue morphology within ovarioles varies across insects, and has been classified into several modes of oogenesis based on the presence and position of special oocyte-associated cells called nurse cells [7].

Here, we compiled 3355 records of ovariole number from across 28 orders, 301 families and 2103 species of insects. We combined these data with published datasets of egg size [9], fecundity [10,11] and body size [12], to test hypotheses about the evolutionary trade-off between offspring size and number. In these analyses, we used an existing phylogeny of insects [13] to analyse evolutionary patterns in ovariole number, and found that hypotheses about life-history evolution do not hold generally true across insects. We then combined these data with published observations of the mode of oogenesis [7], and reconstructed the evolutionary history of the presence and position of nurse cells that contribute to the oocyte during oogenesis. We tested whether patterns in the distribution of ovariole number, egg size or egg shape were driven by the evolution of nurse cells, and found no significant results.

Instead, we observed that the phylogenetic distribution of ovariole number suggests a model where the developmental mechanisms that govern ovariole number have shifted between variable and invariant states several times over the course of insect evolution. Based on this finding, we propose that the developmental mechanisms used to establish ovariole number in well-studied insects such as *Drosophila melanogaster* are unlikely to regulate ovariole number in all insects.

## 2. Methods

### (a) Gathering trait data

We searched the published literature for references to insect ovariole number using a predetermined set of 131 search terms, entered into Google Scholar (scholar.google.com) between June and October of 2019. Each search term comprises an insect taxonomic group and the words 'ovariole number'. The taxonomic groups used in the search process included all insect orders, many large insect families, and taxonomic groups that are well represented in the insect egg dataset [9]. For each Google Scholar search, we evaluated the first 10 publications in the search results. For 61 search terms that had a large number of informative hits, significant representation in the egg dataset, or that corresponded to very speciose groups, we evaluated an additional 20 publications. The list of search terms is available in the electronic supplementary file 'ovariole_number_search_ terms.tsv'.

Using this approach, we gathered 3355 records for ovariole number from 28 insect orders, 301 families and 2103 species, using 448 publications that are listed in the electronic supplementary file 'ovariole_number_bibliography.pdf'. We matched these records to additional taxonomic information using the software TaxReformer [14]. For all subsequent analyses, we excluded observations made in non-reproductive individuals from eusocial species (e.g. workers), as well as two observations that represented significant outliers and could not be validated using additional sources [15,16]. See the electronic supplementary material, methods §1 for details.

For records of ovariole number that reported intraspecific variation in ovariole number, we calculated the per cent difference as follows: if ovariole number was reported as a range, per cent difference was calculated as $100 \times ((\text{max} - \text{min})/\text{median})$; if ovariole number was reported as an average with deviations, per cent difference was calculated as $100 \times ((2 \times \text{deviation})/\text{mean})$. When independent observations of ovariole number for a given species were available from multiple published records, we calculated the per cent difference as $100 \times ((\text{max} - \text{min})/\text{median})$.

We combined the data we collected on total ovariole number with existing datasets of egg size and shape [9], insect lifetime fecundity and dry adult body mass [10,11,17], average adult body length per insect family [12], several lineage-specific measures of adult body size [18–22], and the mode of oogenesis [7]. See the electronic supplementary material, methods §3.1 for details.

All continuous traits (ovariole number, egg volume, lifetime fecundity and all measures of body size) were $\log_{10}$ transformed for subsequent analyses.

## (b) Phylogenetic analyses

The analyses in this manuscript were performed using the insect phylogeny published in Church *et al.* [13], unless otherwise specified. For regressions involving body size data that were reported as insect family-level averages, we used the insect phylogeny published in Rainford *et al.* [23]. Analyses of Drosophilidae ovariole number, egg size and body size were performed using a phylogeny newly assembled for this study. See the electronic supplementary material, methods §2 for details.

To evaluate the robustness of our results to uncertainty in the phylogenetic relationships, all phylogenetic generalized least squares (PGLS) analyses were performed 1000 times over a posterior distribution of trees, using a Brownian Motion based covariance matrix in the R package ape (v. 5.4.1) [24] and nlme (v. 3.1.151) [25]. For regressions at the species and genus level, we reshuffled and matched records for each iteration to account for variation across records for the same taxon. For regressions at the family level, we recalculated the average ovariole number per insect family, randomly downsampling the representation for each family by half. To weight traits by body size, we calculated the phylogenetic residuals [26] of each trait to body size, and then compared the evolution of these residuals using a PGLS regression. See the electronic supplementary material, methods §3.2 for details.

For two regressions comparing egg size to ovariole number while accounting for adult body size, we tested alternative hypotheses of evolution by simulating new data. We considered two such hypotheses: no evolutionary correlation with ovariole number, and a strong correlation with ovariole number (slope of −1). For each trait, we simulated 1000 datasets using evolutionary parameters fitted under a Brownian Motion model in the R packages geiger (v. 2.0.7) [27], and phylolm (v. 2.6.2) [28].

Ancestral state reconstruction of oogenesis mode was performed with the R package corHMM (v. 1.22) [29], and models of trait evolution were compared using the R package Ouwie (v. 1.57) [30]. Ancestral state reconstruction and model comparison were repeated 100 times over a posterior distribution of trees

and resampling data to account for variation across records for the same taxon. See the electronic supplementary material, methods §4.3.

Other comparisons of model fit were performed using the R package geiger (v. 2.0.7) [27] and validated using a parametric bootstrap with the R package arbutus (v. 0.1) [31]. See the electronic supplementary material, methods §5.1.

Analyses of evolutionary rate were performed using BAMM (v. 2.5.0) [32]. For this analysis, we calculated the average ovariole number ($\log_{10}$ transformed) for each genus present in the phylogeny (507 taxa). We used the R package BAMMtools (v. 2.1.7) [33] to select appropriate priors, and ran BAMM for the maximum number of generations ($2 \times 10^{-9}$), sampling every $10^{6}$ generations. Convergence was evaluated both visually (electronic supplementary material, figure S12) and numerically. Running BAMM for the maximum possible number of generations and selecting the optimum burn-in (electronic supplementary material, figure S13) resulted in an effective size for the number of shifts of 482.51, and for the log-likelihood of 149.15. Repeated BAMM analyses showed similar distributions of high and low rate regimes, indicating the implications for ovariole number evolution are robust to uncertainty in rate estimates. See the electronic supplementary material, methods §5.2 for details.

We visualized the results from the BAMM analysis to establish a threshold ($10^{-4}$) for assigning a binary rate regime to each node in the phylogeny, categorizing them as above ('variable') or below ('invariant') a threshold that separates these two peaks.

## (c) Statistical significance

All phylogenetic regressions were performed using the maximum clade credibility tree (the tree with highest credibility score from the posterior distribution of the Bayesian analysis). We considered a relationship significant when the *p*-value was below the threshold 0.01. To assess the robustness of results to uncertainty in phylogenetic relationships, we also repeated these analyses over the posterior distribution of phylogenetic trees and report the number of regressions that gave a significant result (see the electronic supplementary material, table S1).

For two comparisons, we validated that our tests had sufficient statistical power using the selected threshold by comparing the distribution of *p*-values from regressions of observed data to regressions of data simulated under alternative hypotheses. We compared the results of analyses of our observed data to those based on simulated data to evaluate the likelihood of false positives (comparing to data simulated under no correlation) and false negatives (comparing to data simulated with strong correlation).

Model comparisons of trait evolution were also performed over a posterior distribution and accounting for phenotypic uncertainty. For these analyses, we considered a model to have significantly better fitted the data than other models when the difference in the corrected Akaike information criterion (AICc) was greater than two in every analysis iteration.

## 3. Results

## (a) Ovariole number diversity

Ovariole number varies by at least four orders of magnitude across insect species (figure 1*b*). We identified seven insect families with species that have been reported to have more than 1000 total ovarioles, including several eusocial insects (e.g. queens of the termite species *Hypotermes obscuriceps*, Blattodea: Termitidae [34], and several ant species, Hymenoptera: Formicidae) [35,36] and non-eusocial insects (e.g. the blister beetle *Meloe proscarabaeus*, Coleoptera: Meloidae) [37]. We also find two independent lineages that have evolved to have

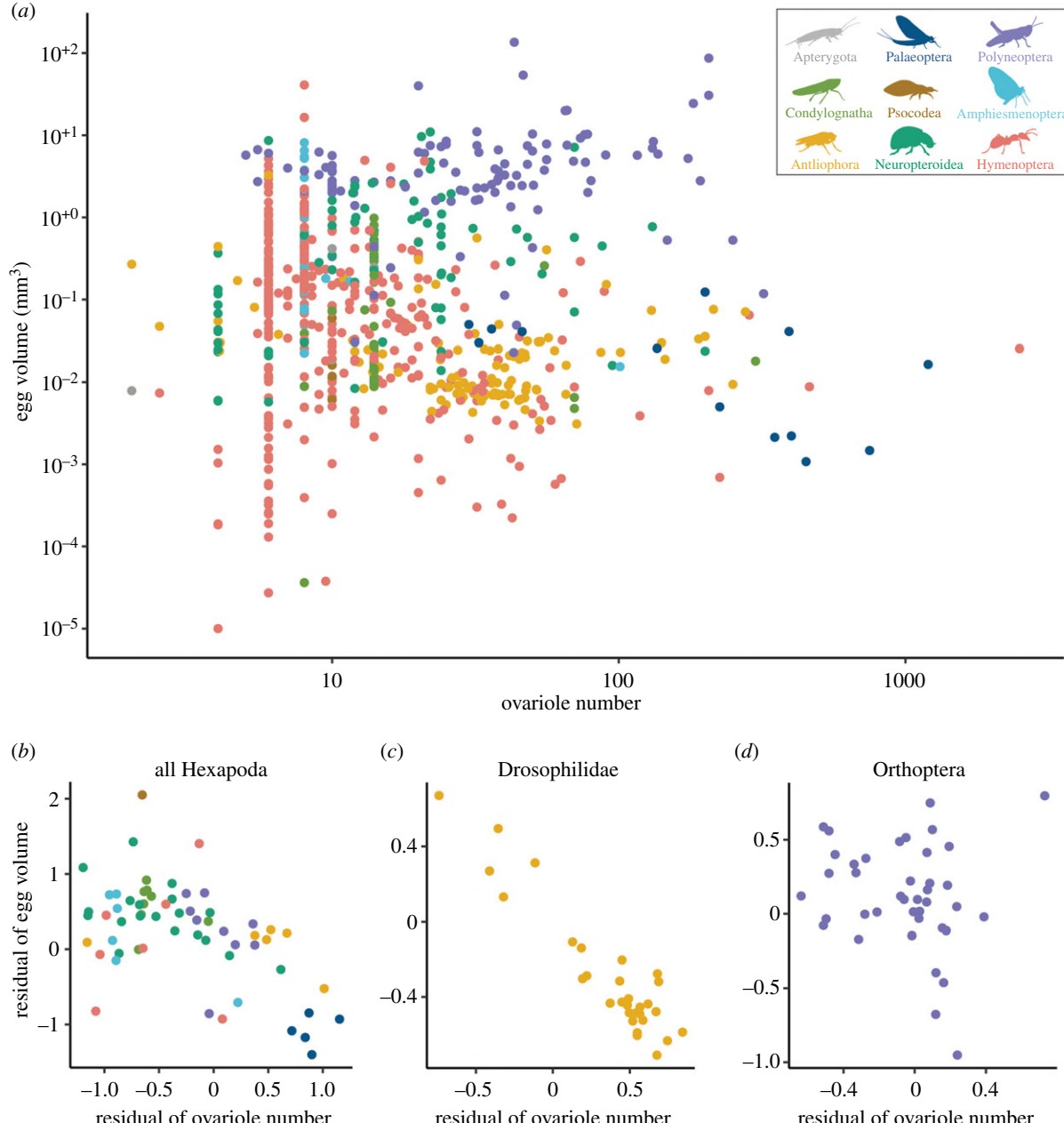

**Figure 2.** Tests of the hypothesized trade-off between egg size and ovariole number. (*a*) Egg volume (mm³) and ovariole number, both log₁₀ scale; points represent insect species. See §3e, Modelling ovariole number evolution for discussion of the enrichment of certain low values of ovariole number (i.e. points appearing vertically arranged). (*b*) Egg volume and ovariole number, residuals to dry adult body mass, points represent genera. (*c*) Drosophilidae egg volume and ovariole number, residuals to thorax length, points represent species. (*d*) Orthoptera egg volume and ovariole number, residuals to body length, points represent genera.

only one functional ovariole: dung beetles in the tribe Scarabaeinae (Coleoptera: Scarabaeidae) [38], and grass flies in the genus *Pachylophus* (Diptera: Chloropidae) [39,40]. In these insects, one of the two ovaries presumably established during embryogenesis is reported to atrophy during development [40,41], resulting in an asymmetric adult reproductive system. We also evaluated intraspecific variation in ovariole number, and found that, for species for which it has been reported, the average per cent difference number within species is between 10% and 100% of the median value (electronic supplementary material, figure S1).

## (b) Ovariole number, egg size and body size

Ovariole number has been hypothesized to be negatively correlated with egg size [5,21,42]. This hypothesis is based on the predictions that (i) female reproduction is resource-limited, therefore egg size should trade off with egg number, and (ii) ovariole number can serve as a proxy for egg number [2,42]. We did not observe a significant negative relationship when comparing egg size and ovariole number across insect species (figure 2*a*; electronic supplementary material, table S1; *p*-value 0.195, *n* = 306). We also compared egg size and ovariole number, combining data from species within the same genus to increase sample size, and again did not observe a significant relationship (electronic supplementary material, figure S2; *p*-value 0.066, *n* = 482). To verify this finding was not driven by the high ovariole numbers seen in the queens of some eusocial insects, we repeated this comparison excluding insects from families with eusocial representatives, with the same result (electronic supplementary material, figure S3; *p*-value 0.209, *n* = 415).

Given that this predicted relationship is often conditioned on body size, which is predicted to limit total potential reproductive

investment [21,43], we combined data on ovariole number and egg size with data on insect adult body mass [10,11,17] and length [12]. When accounting for adult body mass, we observed a significant negative relationship between egg size and ovariole number across genera and species (figure 2b; electronic supplementary material, figure S4; p-value 0.003, slope −0.399, n = 61). To evaluate the robustness of this result, we repeated the analysis 1000 times, taking into account uncertainty in both the phylogeny and trait measurements. Out of 1000 regressions, 995 indicated a significant negative relationship (electronic supplementary material, table S1). We performed the same comparison accounting for adult body length, and likewise observed a significant negative relationship (electronic supplementary material, figure S5; p-value < 0.001, slope −0.52, n = 126), supported by 966 of 1000 repeated analyses (electronic supplementary material, table S1).

We further explored these results using two methods. First, to evaluate our findings against alternative evolutionary hypotheses, we compared these results to regressions based on simulated data. Our results showed that when considering body size, the slope of the regression of egg size and ovariole number is more negative than we would expect to observe by chance, as assessed by comparing to data simulated with no evolutionary correlation (electronic supplementary material, figure S6). However, for both adult body length and dry mass, the slope of the regressions on observed data are not within the range that would be expected under a strong negative correlation (slope of −1 in log-log space; electronic supplementary material, figure S6). This suggests the presence of a weak evolutionary relationship between ovariole number and egg size, when accounting for body size.

Second, we assessed the relationship between egg size and ovariole number, accounting for body size, within four subclades of insects. We found that across Drosophilidae fly species, egg size is indeed strongly negatively correlated with ovariole number when accounting for body size (figure 2c; electronic supplementary material, table S2; p-value < 0.001, slope −0.809, n = 30). By contrast, across grasshoppers and crickets (Orthoptera), beetles (Coleoptera) and wasps (Hymenoptera), we observed no significant relationship between ovariole number and egg size, even when accounting for body size (figure 2d; electronic supplementary material, figure S7 and table S2; Orthoptera: p-value 0.485, n = 40; Coleoptera: p-value 0.384, n = 30; Hymenoptera: p-value 0.139, n = 21). This indicates that, while a strong negative correlation between egg size and ovariole number exists for some insects, it does not represent a universal pattern across insect clades.

Finally, we tested whether ovariole number is positively correlated with adult body size, and in contrast to previous studies [4], we found no correlation between ovariole number and adult body mass or length across insects (electronic supplementary material, figure S8 and table S3; body mass: p-value 0.618, n = 61; body length: p-value 0.031, n = 98). Of the four subclades considered, only insects in the order Orthoptera had a positive relationship between body size and ovariole number (electronic supplementary material, table S3; p-value 0.001, slope 0.35, n = 40).

## (c) Ovariole number and fecundity

If the hypothesized trade-off between the number and size of offspring is true for insects, then one explanation for the lack of a consistent negative relationship between ovariole

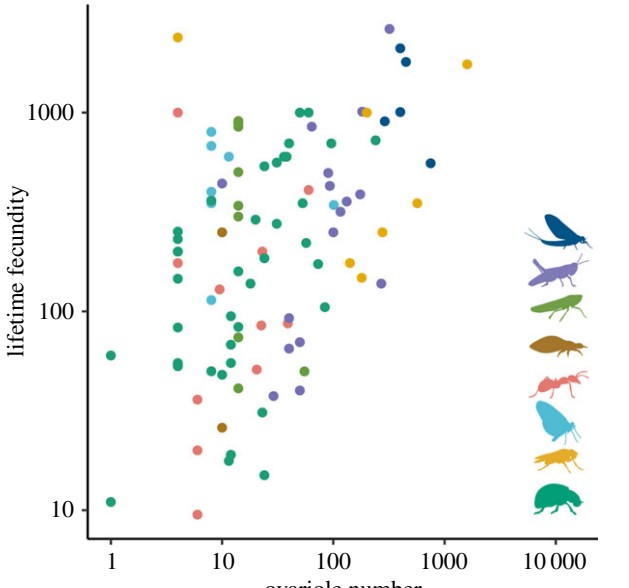

**Figure 3.** The relationship between lifetime fecundity and ovariole number. Both values are shown on a log₁₀ scale. Points represent insect genera and are coloured according to the groups shown in figure 1b.

number and egg size is that ovariole number may not be a reasonable proxy for offspring number. Previous research has shown that, across individuals within the same species, ovariole number is correlated with certain measurements of fecundity, such as maximum daily rate of egg production for *Drosophila* [44,45], but not others, such as lifetime fecundity [46] or fitness in competition assays [47]. Few studies have compared fecundity and ovariole number across species [43], probably owing to the difficulties of measuring fecundity consistently across insects, many of which lay eggs singly and continuously rather than in distinct clutches.

Using a previously reported dataset of lifetime fecundity measurements across insects [10,11], we assessed the relationship between lifetime fecundity and ovariole number. We observed a significant positive relationship (figure 3, p-value 0.002, slope 1.233, n = 65); however, a substantial fraction of repeat analyses show these results are not robust to uncertainty (733 of 1000 regressions are not significant; electronic supplementary material, table S4). We note that this relationship is largely defined by the absence of insects with high ovariole number and low fecundity (figure 3, empty bottom right corner), while for insects with low ovariole number, fecundity varied over more than three orders of magnitude. We interpret our results, in conjunction with those previously reported, to suggest that ovariole number, when variable across insects in a lineage, may be one factor among many influencing the number of eggs produced. However, we caution against using ovariole number as a direct mathematical proxy for offspring number.

## (d) Evolution of nurse cells

In addition to the number of ovarioles, insect ovary morphology has been classified into several modes of oogenesis based on the presence and position of cells that provide nutritive and patterning molecules to the oocyte, which are called nurse cells [7] (figure 4a). Egg formation in the well-studied species *D. melanogaster* is an example of a meroistic oogenesis

**Figure 4.** The evolution of the presence and position of nurse cells. (*a*) Insect oogenesis was categorized into several modes by Büning [7] based on the presence and position of nurse cells. (*b*) Phylogenetic reconstruction of mode of oogenesis. Scale bar indicates 100 million years (Myr). Grey, panoistic ovaries, without nurse cells; cyan, polytrophic meroistic ovaries, with nurse cells adjacent to maturing oocytes; red, telotrophic meroistic ovaries, with nurse cells located in germaria; black, unique meroistic ovary type observed in Strepsiptera. Insect taxonomic groups are, from top to bottom: Apterygota, Palaeoptera, Polyneoptera, Condylognatha, Psocodea, Hymenoptera, Neuropteroidea, Amphiesmenoptera and Antliophora.

mode, meaning that its ovarioles contain nurse cells of germ line origin that are connected to developing oocytes via cytoplasmic bridges [48]. In insects with a polytrophic meroistic arrangement, these nurse cells are clonally related and immediately adjacent to each oocyte. An alternative arrangement is seen in telotrophic meroistic ovaries, where oocytes in each ovariole are connected to a common pool of nurse cells located in the germarium [7]. Meroistic ovaries are thought to have evolved from an ancestral panoistic mode, meaning they lack nurse cells [7]. Using a previously published set of descriptions of these oogenesis modes across insects [7], we reconstructed the evolutionary transitions between these states. Consistent with previous analyses [7], we found that the ancestral insect probably had panoistic ovaries (lacking nurse cells), with several independent shifts to both telotrophic and polytrophic meroistic modes, and at least two reversals from meroistic back to panoistic (figure 4*b*; electronic supplementary material, figure S10).

Using this ancestral state reconstruction, we then compared models of trait evolution to test whether evolutionary transitions in oogenesis mode helped explain the diversification of ovariole number and egg morphology. We found that, for the traits studied here, models that take into account evolutionary changes in mode of oogenesis do not consistently demonstrate a significant improvement over models that do not take these changes into account ($\Delta$AIC < 2; electronic supplementary material, table S5). In other words, the evolution of nurse cells and their position within the ovary do not explain the diversification of egg size, egg shape, or ovariole number.

To analyse the robustness of these results to uncertainty in the tree topology and in the inference of ancestral states, we repeated each analysis over a posterior distribution of trees. For egg asymmetry and curvature, but not for volume or aspect ratio, we observed a few iterations where a model that takes into account oogenesis mode evolution was significantly favoured over models that did not ($\Delta$AIC > 2;

electronic supplementary material, table S5). However, this result was infrequent over 100 repetitions of the analysis. We, therefore, interpret these results as suggestive of a possible relationship between mode of oogenesis and egg asymmetry and curvature, but one which cannot be confirmed given the current data available.

## (e) Modelling ovariole number evolution

Using the dataset compiled here and a previously published phylogeny of insects (figure 5*a*) [13], we modelled the rate of evolutionary change in ovariole number (electronic supplementary material, figures S11–S14). We observed substantial rate heterogeneity in the evolution of ovariole number (electronic supplementary material, figure S14), meaning that for some lineages ovariole number has evolved rapidly where in others, ovariole number has evolved very slowly or not at all. The most striking example of this is the multiple lineages which have independently evolved invariant or near-invariant ovariole number across taxa (e.g. nearly all Lepidoptera have exactly eight ovarioles, figure 5*b*; Lepidoptera are part of Amphiesmenoptera, in cyan), from an ancestral variable state. These invariant lineages were identified by finding regions of the phylogeny that experience extremely low rates of ovariole number diversification (electronic supplementary material, figures S14 and S15). Using this approach, we found that invariant ovariole numbers have evolved at least nine times independently across insects, with several subsequent reversals from invariant to variable states (figure 5*a*).

We found that the rate of evolutionary change in ovariole number is correlated with the number of ovarioles: lineages with relatively low ovariole number also experience relatively low degrees of ovariole number change (electronic supplementary material, figure S11). This is evidenced by the fact that, of the nine invariant lineages, none have greater than seven ovarioles per ovary (figure 5*c*). However, we note that not all insects with low ovariole counts are in invariant lineages; many insects with fewer than 14 total ovarioles are in lineages with relatively high rates of intra- and interspecific ovariole number variation (figure 5)

The distribution of ovariole numbers across insects is enriched for even numbers of total ovarioles (figure 5*c*). While many insects show asymmetries in the number of ovarioles between the left and right ovaries, all of the invariant lineages are symmetric (at 4, 6, 8, 10, 12 and 14 total ovarioles). Additionally, for the insects identified as part of invariant lineages, none have any reported intraspecific variation in ovariole number. Therefore, invariant lineages have near-zero variation when comparing between species, between individuals within a species, and between the left and right ovary within an individual.

Using these results, we propose a multi-rate model, where the rate of ovariole number evolution differs based on the evolution of a discrete trait representing invariant or variable status. We propose that the evolution of this discrete trait is governed by a model where the likelihood of transitions from a variable to an invariant state is negatively correlated with the current number of ovarioles. Here, we demonstrate that a multi-rate Brownian motion model far outperforms a single rate model in fitting the data ($\Delta$AICc 1770.93). In addition, using a parametric bootstrap to evaluate model fit, we find evidence that processes beyond Brownian motion processes are probably at play (electronic

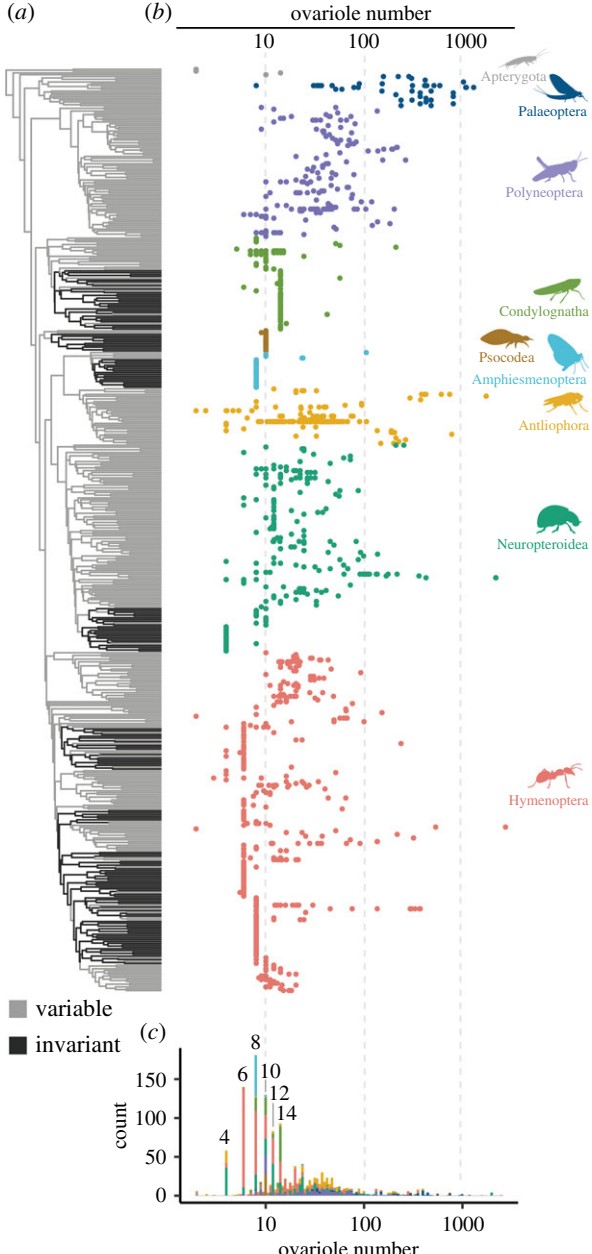

**Figure 5.** The evolutionary distribution of ovariole number across insects. (*a*) Phylogeny of insect genera, coloured according to the inferred rate regime of ovariole number evolution, variable in grey and invariant in black (see the electronic supplementary material, Methods). (*b*) Total ovariole numbers, shown on a log$_{10}$ scale and arranged by insect genus according to the phylogeny. Tips with more than one point represent genera with multiple records for total ovariole number in the dataset. (*c*) The distribution of values shown in (*b*), showing enrichment for even values in the left tail of the distribution.

supplementary material, figure S11) [31]. We suggest that as researchers continue to develop non-Gaussian models for continuous trait evolution [49], those models will be useful for describing the evolution of ovariole number.

## 4. Discussion

A frequently invoked life-history prediction is that, given a finite set of metabolic resources, organisms can either produce few offspring, each with high fitness, or many low-fitness offspring [1–3]. In insects, egg size and ovariole number are often used as proxies for offspring fitness [50] and number [44,45], respectively, and therefore it has been predicted that insects with more ovarioles lay smaller eggs than insects with fewer ovarioles [5,6,21,42]. Our results, using a dataset that spans 3355 observations across 2103 species, and that takes into account phylogenetic relationships, indicate that a generalized trade-off between insect egg size and ovariole number does not exist (figure 2).

Lineages of insects with invariant ovariole number illustrate this point. Despite having the same ovariole number, these lineages contain a range of egg sizes that is comparable to the four orders of magnitude observed across all insects (figure 2*a*). Furthermore, we observed no relationship between the evolutionary rates of change for ovariole number and egg size (electronic supplementary material, figure S17). Therefore, if a trade-off between egg size and fecundity exists, factors beyond variation in ovariole number must contribute to fecundity. These factors might include variation in the rate of egg production per ovariole [51–54], among others [55,56].

We suggest that considering the evolution of developmental processes that govern ovariole number specification may be more useful in explaining patterns of diversity than predictions based on metabolic trade-offs. As evidence of this, we point to the fact that invariant lineages appear to have near-zero variation not only across species, but also within species, and between the left and right ovary within individuals. This suggests that the mechanism which determines ovariole number has become canalized in these groups. By contrast, much of the existing research on how ovariole number is regulated has studied *D. melanogaster*, where the number of ovarioles can vary between the left and right ovaries within an individual, as well as across individuals within a population [57,58]. In this species, adult ovariole number is determined by cell proliferation and rearrangement during larval development [59,60]. Variation in adult number is derived primarily from variation in the number of 'terminal filament precursor cells' [61,62], as well as from variation in the number of those precursor cells that group together to form the structure that initiates ovariole formation, known as a 'terminal filament' [63]. Across species of *Drosophila*, variation in average adult ovariole number results primarily from variation in the average number of terminal filament precursor cells [62].

When considering the developmental processes that could give rise to invariant ovariole number, we propose that the major determinants of ovariole number known from *Drosophila* may not apply. To achieve an invariant ovariole number, these processes might instead include mechanisms for strict counting of individual cells or discrete cell subpopulations. In the former, if the cells that ultimately comprised a terminal filament were derived by mitotic division from a single progenitor, rather than by cellular rearrangements as is the case in *Drosophila* [59], then an invariant ovariole number could be achieved via strict control of the number of precursor cells. Alternatively, an invariant ovariole number could be achieved by partitioning the starting population of precursor cells into a tightly regulated number of subpopulations. This would again be a departure from known mechanisms in *Drosophila*, in which a variable number of precursor cells are gathered into terminal filaments until the population is depleted [59,63]. The determining factor for partitioning the precursor pool could be, for example, a spatially variable morphogen emanating from adjacent tissues [64] or a reaction–diffusion patterning process [65] within the developing

ovary, as these have been shown to generate fixed numbers of multicellular structures in other developmental contexts [66–68]. These predictions could be tested by characterizing the dynamics of cell number and position across invariant lineages, and making comparisons to corresponding data from their variable relatives.

The evolutionary transitions between variable and invariant ovariole number are reminiscent of other quantitative traits across multicellular life, including patterns of variability and invariance in arthropod segment number [69,70], vertebrate digit number [71,72], or the number of angiosperm floral organs [73,74]. Across these systems, the evolutionary history of morphogenetic counting mechanisms is poorly understood. We suggest that insect ovariole number presents an ideal case to study this phenomenon. In particular, we note the evidence that invariance has evolved convergently at least nine times, as well as the evidence of several reversals back to variability from an invariant ancestral state (figure 5). These convergent lineages provide an opportunity to test the predictability of evolutionary changes to counting mechanisms, by asking whether convergent evolution of invariance involves convergent canalization of shared molecular mechanisms.

Data accessibility. The dataset of insect ovariole number is available from the Dryad Digital Repository: https://dx.doi.org/10.5061/dryad. 59zw3r253 [75]. The code and phylogenetic trees required to reproduce all the analyses, figures, and generate the manuscript files are provided at https://github.com/shchurch/insect_ovariole_number_evolution_ 2020, commit 6cf446a. All analyses performed in R (v. 4.0.3) were done so in a clean environment, built with conda (v. 4.9.2), and instructions for rebuilding this environment are provided in the same repository.

Authors' Contributions. S.H.C. led the data collection, analysis and writing of the manuscript. N.M. performed the initial literature search, data collection and analysis. B.A.S.d.M. assembled all phylogenies for analysis. S.D., B.A.S.d.M. and C.G.E. contributed to data analysis, visualization, discussion and writing. All authors gave final approval for publication and agreed to be held accountable for the work performed therein.

Competing interests. The authors declare no competing interests.

Funding. This work was supported by NSF GRFP DGE1745303 to S.H.C. and funds from Harvard University to support S.H.C. and C.G.E. N.M. was supported by the E3 REU program at Harvard University.

Acknowledgements. We thank members of the Extavour Laboratory for discussion.

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
