## [Peer Review File · Proceedings of the Royal Society B: Biological Sciences]

Review History

RSPB-2021-0150.R0 (Original submission)

Review form: Reviewer 1 (James Gilbert)

Recommendation

Accept with minor revision (please list in comments)

Scientific importance: Is the manuscript an original and important contribution to its field?

Excellent

General interest: Is the paper of sufficient general interest?

Excellent

Quality of the paper: Is the overall quality of the paper suitable?

Excellent

Is the length of the paper justified?

Yes

Should the paper be seen by a specialist statistical reviewer?

No

Do you have any concerns about statistical analyses in this paper? If so, please specify them explicitly in your report.

No

It is a condition of publication that authors make their supporting data, code and materials available - either as supplementary material or hosted in an external repository. Please rate, if applicable, the supporting data on the following criteria.

Is it accessible?

Yes

Is it clear?

Yes

Is it adequate?

Yes

Do you have any ethical concerns with this paper?

No

Comments to the Author

I think this is a wonderful paper that is unquestionably suitable for publication, subject to a couple of minor suggestions which I detail below. It is of broad appeal to the readers of Proceedings B as it deals with fundamental questions of reproductive allocation, evolution, tempo and mode, and more, for a large taxonomic group of wide appeal to researchers. The authors have compiled a large set of published data on insect ovariole numbers alongside various other published life history traits, and test hypotheses about the evolution of reproductive allocation and reproductive mode. Some of the results are (to me) astonishing, and are likely to stimulate a large number of new hypotheses and research. I would therefore recommend acceptance with minor revisions.

The methods to me appear robust, although I note that I'm not a methodological expert here; I'm a user, rather than a designer of such methods (and I have very limited experience with analyses of evolutionary rate). I would note that it is possible to account for variation within species in a phylogenetic regression within one analysis by using Bayesian mixed models in the MCMCglmm package, rather than using multiple GLS analyses and picking randomly among the data each time. Given that the authors have used Bayesian methods elsewhere in the MS, this might be a suggestion to consider. Nevertheless, I have no inherent problem with the data-shuffling method that the authors have used here. Finally, I also note that the data the authors use from their ref 10, a PhD thesis, are now published (Gilbert & Manica, 2010, American Naturalist; Gilbert 2011, Florida Entomologist), so these may be more appropriate citations here.

The finding that ovariole number and egg volume are not associated unless you account for body size is very interesting. Looking at figure 2a, though, I wonder whether these traits would be associated if you excluded the eusocial Hymenoptera and termites. I think you could make a good case for excluding eusocial queens given that reproductive trade-offs in these groups may be affected by the division of labour seen in eusocial colonies. More broadly on this point (trade-offs against reproductive investment), I think this finding suggests further hypotheses to do with post-oviposition investment in offspring - whether the relationship between ovariole number and egg size might depend upon investment in each offspring by the parents - maybe a route for further research.

The finding about the evolution of variable versus invariant numbers of ovarioles is fantastically interesting. The authors suggest that "if a trade-off between egg size and fecundity exists, factors beyond variation in ovariole number must contribute to fecundity." I think a pertinent question

here might possibly be whether ovariole number *mediates* a trade-off between egg size and lifetime fecundity, i.e. whether you'd see that tradeoff within lineages that carry invariant ovariole numbers (or whether variable numbers somewhat obscure the tradeoff). To this end, I think it might have been a good idea to include multiple regression analyses here with body size/lifetime fecundity included as well as just egg size and ovariole number, rather than a successive set of bivariate ones – I think a number of revealing interactions might have been detectable with the large sample sizes the authors have employed.

MINOR POINTS

Line 148: please define what you mean by maximum clade credibility tree

Line 157: simulated

Line 204-5 "the slope of the regression is lower" - please clarify whether you mean "lower in magnitude", or "more negative" here.

Line 224 There is no number for the figure or table specified.

Line 310-4. Please indicate the position of the Lepidoptera within the Neuropteroidea on the figure.

Figure 5. I noticed there is one group within the Neuropteroidea which has apparently near-invariant ovariole numbers (or at least few that vary) but which do not appear in black – is this worth mentioning or explaining?

Review form: Reviewer 2

Recommendation

Major revision is needed (please make suggestions in comments)

Scientific importance: Is the manuscript an original and important contribution to its field?

Excellent

General interest: Is the paper of sufficient general interest?

Excellent

Quality of the paper: Is the overall quality of the paper suitable?

Good

Is the length of the paper justified?

Yes

Should the paper be seen by a specialist statistical reviewer?

Yes

Do you have any concerns about statistical analyses in this paper? If so, please specify them explicitly in your report.

No

It is a condition of publication that authors make their supporting data, code and materials available - either as supplementary material or hosted in an external repository. Please rate, if applicable, the supporting data on the following criteria.

Is it accessible?

Yes

Is it clear?

Yes

Is it adequate?

Yes

Do you have any ethical concerns with this paper?

No

Comments to the Author

The main aim of “Repeated loss of variation in insect ovary morphology highlights the role of developmental constraint in life- history evolution” by Church et al. is to test a well-known life history prediction, namely that there is a trade off between the number and size of offspring using developmental traits (ovariole number and egg size). The authors use an impressive comparative database that they have compiled from the literature and use sophisticated phylogenetic comparative methods to test this hypothesis. They conclude that the hypothesized relationship is not general across insects, but rather a characteristic of particular groups, like the Drosophilidae. They also explore the relationship between nurse cell arrangement and egg size, which was also insignificant, as well as the dynamics of ovariole number evolution across insects. The most interesting result is that ovariole number is significantly variable in some groups and invariant in others, and that the invariant groups evolved several times independently. This last result, in our view, will be of interest to the broad readership of PRS-B and we recommend its publication. However, we would like the Authors to address the following comments and critiques to help improve the clarity and quality of the manuscript:

MAJOR COMMENTS:

(1) In their analyses, Church et al. control for body size by calculating phylogenetically controlled residuals of egg size versus body size and residuals of ovariole number versus body size. They then use these residuals to assess the relationship between ovariole number versus body-size. The reason why body-size is frequently “controlled for” is to ensure that the correlation between two variables of interest (in this case ovariole number and egg size) is not due to a third variable (in this case body size). However, at the beginning of the manuscript, the reader is left to assume that there exists a significant relationship between body size and ovariole number and body size and egg size, which must be accounted for. But on lines 222 to 227 (and figure S5), the reader surprisingly learns that there is in fact no significant relationship between ovariole number and body size (with the exception of orthopterans). This leaves that reader wondering about whether there is in fact a significant relationship between egg size and body size, which is actually never shown or reported. Therefore, it becomes unclear whether it is actually necessary to control for body size and using residuals in the first place.

Furthermore, demonstrating whether there is (or is not) a relationship between egg size and body size is a key consideration because from our humble understanding of the trade-off between offspring number and offspring size, is that offspring size is assumed to be correlated to eventual adult size. So, the tradeoff as we understand it would be as follows: the more offspring, the less energy available to invest in the offspring and so the offspring develop into smaller adults (and vice versa). However, if there is no relationship between egg size and adult size, then egg size is not a good proxy of eventual adult size, and is therefore an indicator of another ecological trait that is independent of size in general. If there is a significant indicator of eventual

adult size, then we find it confusing why we would ‘control’ for body size in this case when it is the very thing the tradeoff is trying to assess. Perhaps we have missed something in our logic, so it would be very helpful if this was made more clear in the manuscript and laid out more clearly both in the introduction and the methods.

(2) In some panels, the Authors represent species-level regressions, while in others they present genus-level regressions, and in the main text (lines 104 to 107) they refer to family-level regressions. There are two points for consideration: the first is that the figures should be better labeled as to not confuse readers at which level the regressions are being performed, and the second (more important) consideration is that in the phylogenetic comparative method literature, there is an important pattern they call the “taxon-level affect”, which is discussed extensively in Harvey and Pagel (1991) and in the brain to body scaling literature. Therefore, the authors should be aware of the taxon-level effect, discuss it, and perhaps try to explain why at the genus level there appears to be a relationship but that the species-level there is not. This result was dismissed by the Authors as not significant, but it may be more telling than the Authors may realize.

(3) In the section on “Modeling ovariole number evolution” the Authors mainly evaluate a Brownian Motion model and conclude that “a multi-rate Brownian motion model far outperforms a single rate model in fitting the data...” While this is entirely valid, we feel that the analyses and results would be more compelling if the authors considered other models of evolutionary change to explain the existence of variable and invariant insect lineages.

(4) The title of the manuscript states that the Authors’ study “highlights the role of developmental constraint in life-history evolution”. However there is no mention of the term ‘developmental constraint’ within the manuscript. It would benefit the discussion to define the concept of ‘developmental constraint’ and contextualise its place in the literature on evolutionary theory. Particularly, the Authors’ finding that low-variation lineages tend to have little variation in ovariole number not just between species, but also within species and between the two ovaries of an individual, is very interesting and warrants further discussion, perhaps in the context of an expanded discussion of developmental constraint. For instance, do the authors view a lack of variation in intraspecific and within-individual ovariole number as evidence of more robust/less plastic ovary development in these species (perhaps due to developmental mechanisms different from the *Drosophila* paradigm, as they discuss)? How do the authors propose that differences in ovary development might drive or bias evolution of ovariole number across species?

MINOR COMMENTS:

Introduction:

(1) On lines 50 to 51, the Authors state that there is both intra- and interspecific variation of ovariole number across species. However, one of the main assumptions for phylogenetic comparative methods (which don’t explicitly take into account intraspecific variation) is that interspecific variation is significantly greater than intraspecific variation. Therefore, the Authors should make this very clear.

(2) The Introduction is quite short. Perhaps Authors would like to mention / discuss whether there are other studies that have used developmental traits to assess the trade off between offspring number and size, and mention what the consensus is based on previous studies. Is this the first test? It would be useful for the reader to situate the results of this study in the context of previous work.

Methods:

(1) Lines 73 to 74 is unclear, perhaps rearrange order of sentence.

(2) In line 148, ‘both’ seems to be a typo.

(3) It is not clear how the Authors dealt with data on intra-specific ovariole number variation. In the methods there is no explicit mention of generating measures of intra-specific variation, nor is such data presented in any of the figures, but in lines 329-331 of the results the Authors state that “invariant lineages have near-zero variation when comparing between species, between individuals within a species, and between the left and right ovary within an individual.” The Authors should point to the data that support their claim of lower within-species variation in these lineages.

Results:

(1) Lines 224 to 225 the figure and table number are missing.

(2) The Authors should ensure the link to their dataset is correct (doi:10.5061/dryad.59zw3r253).

Review form: Reviewer 3

Recommendation

Major revision is needed (please make suggestions in comments)

Scientific importance: Is the manuscript an original and important contribution to its field?

Excellent

General interest: Is the paper of sufficient general interest?

Excellent

Quality of the paper: Is the overall quality of the paper suitable?

Good

Is the length of the paper justified?

Yes

Should the paper be seen by a specialist statistical reviewer?

Yes

Do you have any concerns about statistical analyses in this paper? If so, please specify them explicitly in your report.

No

It is a condition of publication that authors make their supporting data, code and materials available - either as supplementary material or hosted in an external repository. Please rate, if applicable, the supporting data on the following criteria.

Is it accessible?

Yes

Is it clear?

Yes

Is it adequate?

Yes

Do you have any ethical concerns with this paper?

No

Comments to the Author

The main aim of “Repeated loss of variation in insect ovary morphology highlights the role of developmental constraint in life-history evolution” by Church et al. is to test a well-known life history prediction, namely that there is a trade off between the number and size of offspring using developmental traits (ovariole number and egg size). The authors use an impressive comparative database that they have compiled from the literature and use sophisticated phylogenetic comparative methods to test this hypothesis. They conclude that the hypothesized relationship is not general across insects, but rather a characteristic of particular groups, like the Drosophilidae. They also explore the relationship between nurse cell arrangement and egg size, which was also insignificant, as well as the dynamics of ovariole number evolution across insects. The most interesting result is that ovariole number is significantly variable in some groups and invariant in others, and that the invariant groups evolved several times independently. This last result, in our view, will be of interest to the broad readership of PRS-B and we recommend its publication. However, we would like the Authors to address the following comments and critiques to help improve the clarity and quality of the manuscript:

MAJOR COMMENTS:

(1) In their analyses, Church et al. control for body size by calculating phylogenetically controlled residuals of egg size versus body size and residuals of ovariole number versus body size. They then use these residuals to assess the relationship between ovariole number versus body-size. The reason why body-size is frequently “controlled for” is to ensure that the correlation between two variables of interest (in this case ovariole number and egg size) is not due to a third variable (in this case body size). However, at the beginning of the manuscript, the reader is left to assume that there exists a significant relationship between body size and ovariole number and body size and egg size, which must be accounted for. But on lines 222 to 227 (and figure S5), the reader surprisingly learns that there is in fact no significant relationship between ovariole number and body size (with the exception of orthopterans). This leaves that reader wondering about whether there is in fact a significant relationship between egg size and body size, which is actually never shown or reported. Therefore, it becomes unclear whether it is actually necessary to control for body size and using residuals in the first place.

Furthermore, demonstrating whether there is (or is not) a relationship between egg size and body size is a key consideration because from our humble understanding of the trade-off between offspring number and offspring size, is that offspring size is assumed to be correlated to eventual adult size. So, the tradeoff as we understand it would be as follows: the more offspring, the less energy available to invest in the offspring and so the offspring develop into smaller adults (and vice versa). However, if there is no relationship between egg size and adult size, then egg size is not a good proxy of eventual adult size, and is therefore an indicator of another ecological trait that is independent of size in general. If there is a significant indicator of eventual adult size, then we find it confusing why we would ‘control’ for body size in this case when it is the very thing the tradeoff is trying to assess. Perhaps we have missed something in our logic, so it would be very helpful if this was made more clear in the manuscript and laid out more clearly both in the introduction and the methods.

(2) In some panels, the Authors represent species-level regressions, while in others they present genus-level regressions, and in the main text (lines 104 to 107) they refer to family-level regressions. There are two points for consideration: the first is that the figures should be better labeled as to not confuse readers at which level the regressions are being performed, and the second (more important) consideration is that in the phylogenetic comparative method literature, there is an important pattern they call the “taxon-level affect”, which is discussed extensively in Harvey and Pagel (1991) and in the brain to body scaling literature. Therefore, the authors should be aware of the taxon-level effect, discuss it, and perhaps try to explain why at the genus level there appears to be a relationship but that the species-level there is not. This result was dismissed by the Authors as not significant, but it may be more telling than the Authors may realize.

(3) In the section on “Modeling ovariole number evolution” the Authors mainly evaluate a

Brownian Motion model and conclude that “a multi-rate Brownian motion model far outperforms a single rate model in fitting the data...” While this is entirely valid, we feel that the analyses and results would be more compelling if the authors considered other models of evolutionary change to explain the existence of variable and invariant insect lineages.

(4) The title of the manuscript states that the Authors’ study “highlights the role of developmental constraint in life-history evolution”. However there is no mention of the term ‘developmental constraint’ within the manuscript. It would benefit the discussion to define the concept of ‘developmental constraint’ and contextualise its place in the literature on evolutionary theory. Particularly, the Authors’ finding that low-variation lineages tend to have little variation in ovariole number not just between species, but also within species and between the two ovaries of an individual, is very interesting and warrants further discussion, perhaps in the context of an expanded discussion of developmental constraint. For instance, do the authors view a lack of variation in intraspecific and within-individual ovariole number as evidence of more robust/less plastic ovary development in these species (perhaps due to developmental mechanisms different from the *Drosophila* paradigm, as they discuss)? How do the authors propose that differences in ovary development might drive or bias evolution of ovariole number across species?

MINOR COMMENTS:

Introduction:

(1) On lines 50 to 51, the Authors state that there is both intra- and interspecific variation of ovariole number across species. However, one of the main assumptions for phylogenetic comparative methods (which don’t explicitly take into account intraspecific variation) is that interspecific variation is significantly greater than intraspecific variation. Therefore, the Authors should make this very clear.

(2) The Introduction is quite short. Perhaps Authors would like to mention / discuss whether there are other studies that have used developmental traits to assess the trade off between offspring number and size, and mention what the consensus is based on previous studies. Is this the first test? It would be useful for the reader to situate the results of this study in the context of previous work.

Methods:

(1) Lines 73 to 74 is unclear, perhaps rearrange order of sentence.

(2) In line 148, ‘both’ seems to be a typo.

(3) It is not clear how the Authors dealt with data on intra-specific ovariole number variation. In the methods there is no explicit mention of generating measures of intra-specific variation, nor is such data presented in any of the figures, but in lines 329-331 of the results the Authors state that “invariant lineages have near-zero variation when comparing between species, between individuals within a species, and between the left and right ovary within an individual.” The Authors should point to the data that support their claim of lower within-species variation in these lineages.

Results:

(1) Lines 224 to 225 the figure and table number are missing.

(2) The Authors should ensure the link to their dataset is correct (doi:10.5061/dryad.59zw3r253).

Decision letter (RSPB-2021-0150.R0)

24-Feb-2021

Dear Dr Church:

Your manuscript has now been peer reviewed and the reviews have been assessed by an Associate Editor. The reviewers' comments (not including confidential comments to the Editor) and the comments from the Associate Editor are included at the end of this email for your reference. As you will see, the reviewers and the Editors have raised some concerns with your manuscript and we would like to invite you to revise your manuscript to address them.

Research ethics:

Use of animals and field studies:

It is a condition of publication that you make available the data and research materials supporting the results in the article. Please see our Data Sharing Policies (<https://royalsociety.org/journals/authors/author-guidelines/#data>). Datasets should be deposited in an appropriate publicly available repository and details of the associated accession number, link or DOI to the datasets must be included in the Data Accessibility section of the

article (<https://royalsociety.org/journals/ethics-policies/data-sharing-mining/>). Reference(s) to datasets should also be included in the reference list of the article with DOIs (where available).

Please submit a copy of your revised paper within three weeks. If we do not hear from you within this time your manuscript will be rejected. If you are unable to meet this deadline please let us know as soon as possible, as we may be able to grant a short extension.

Best wishes,
Professor Gary Carvalho
mailto: proceedingsb@royalsociety.org

Associate Editor

Board Member: 1

Comments to Author:

Two independent sets of reviews have been provided (noting that one is jointly produced from the same lab), which are generally positive and agree that this presents an interesting and novel piece of research, of appeal to readers of Proceedings B. The finding of ovariole number being variable in some groups and invariant in others, with this invariance having evolved independently multiple times, was deemed to be of particular interest. I agree with this consensus, that this analysis presents a powerful test of a classic life history prediction, and generates a number of new theories and possibilities for future work. I also found the figures to be extremely clear and effective.

Both reviews provide a number of detailed and constructive suggestions to improve the manuscript, and I do not repeat these here. In addition to the reviewers' points, I make the following suggestions:

- More clarity on rationale to study the evolution of nurse cells in the Introduction - this forms quite a significant part of the analysis, but only gets one sentence in the introduction (1.53-55). I also noticed that the term 'nurse cells' is not mentioned in the discussion. It likely relates to the point about precursor cells (1.363 onwards), but it would help for non-experts if there was more consistency in terminology. (Note that this point is related to Reviewer 2's general point 4 about more context for 'developmental constraints' mentioned in the title)

- 1.56-59, you could reiterate which 'hypotheses about reproductive evolution' are being tested here.

- I am not familiar with these phylogenetic regressions, but I was surprised that only p-values and n are reported, and no mention of effect size in the main results. I see that slope estimates are given in the supplementary tables, perhaps this is most efficient in terms of space but if there was some room then I would like to these ranges also provided in the main results.

Reviewer(s)' Comments to Author:

Referee: 1

Comments to the Author(s)

I think this is a wonderful paper that is unquestionably suitable for publication, subject to a couple of minor suggestions which I detail below. It is of broad appeal to the readers of Proceedings B as it deals with fundamental questions of reproductive allocation, evolution, tempo and mode, and more, for a large taxonomic group of wide appeal to researchers. The authors have compiled a large set of published data on insect ovariole numbers alongside various other published life history traits, and test hypotheses about the evolution of reproductive allocation and reproductive mode. Some of the results are (to me) astonishing, and are likely to stimulate a large number of new hypotheses and research. I would therefore recommend acceptance with minor revisions.

The methods to me appear robust, although I note that I'm not a methodological expert here; I'm a user, rather than a designer of such methods (and I have very limited experience with analyses of evolutionary rate). I would note that it is possible to account for variation within species in a phylogenetic regression within one analysis by using Bayesian mixed models in the MCMCglmm package, rather than using multiple GLS analyses and picking randomly among the data each time. Given that the authors have used Bayesian methods elsewhere in the MS, this might be a suggestion to consider. Nevertheless, I have no inherent problem with the data-shuffling method that the authors have used here. Finally, I also note that the data the authors use from their ref 10, a PhD thesis, are now published (Gilbert & Manica, 2010, *American Naturalist*; Gilbert 2011, *Florida Entomologist*), so these may be more appropriate citations here.

The finding that ovariole number and egg volume are not associated unless you account for body size is very interesting. Looking at figure 2a, though, I wonder whether these traits would be associated if you excluded the eusocial Hymenoptera and termites. I think you could make a good case for excluding eusocial queens given that reproductive trade-offs in these groups may be affected by the division of labour seen in eusocial colonies. More broadly on this point (trade-offs against reproductive investment), I think this finding suggests further hypotheses to do with post-oviposition investment in offspring - whether the relationship between ovariole number and egg size might depend upon investment in each offspring by the parents - maybe a route for further research.

The finding about the evolution of variable versus invariant numbers of ovarioles is fantastically interesting. The authors suggest that "if a trade-off between egg size and fecundity exists, factors beyond variation in ovariole number must contribute to fecundity." I think a pertinent question here might possibly be whether ovariole number *mediates* a trade-off between egg size and lifetime fecundity, i.e. whether you'd see that tradeoff within lineages that carry invariant ovariole numbers (or whether variable numbers somewhat obscure the tradeoff). To this end, I think it might have been a good idea to include multiple regression analyses here with body size/lifetime fecundity included as well as just egg size and ovariole number, rather than a

successive set of bivariate ones – I think a number of revealing interactions might have been detectable with the large sample sizes the authors have employed.

MINOR POINTS

Line 148: please define what you mean by maximum clade credibility tree

Line 157: simulated

Line 204-5 "the slope of the regression is lower" - please clarify whether you mean "lower in magnitude", or "more negative" here.

Line 224 There is no number for the figure or table specified.

Line 310-4. Please indicate the position of the Lepidoptera within the Neuropteroidea on the figure.

Figure 5. I noticed there is one group within the Neuropteroidea which has apparently near-invariant ovariole numbers (or at least few that vary) but which do not appear in black – is this worth mentioning or explaining?

Referee: 2

Comments to the Author(s)

The main aim of “Repeated loss of variation in insect ovary morphology highlights the role of developmental constraint in life-history evolution” by Church et al. is to test a well-known life history prediction, namely that there is a trade off between the number and size of offspring using developmental traits (ovariole number and egg size). The authors use an impressive comparative database that they have compiled from the literature and use sophisticated phylogenetic comparative methods to test this hypothesis. They conclude that the hypothesized relationship is not general across insects, but rather a characteristic of particular groups, like the Drosophilidae. They also explore the relationship between nurse cell arrangement and egg size, which was also insignificant, as well as the dynamics of ovariole number evolution across insects. The most interesting result is that ovariole number is significantly variable in some groups and invariant in others, and that the invariant groups evolved several times independently. This last result, in our view, will be of interest to the broad readership of PRS-B and we recommend its publication. However, we would like the Authors to address the following comments and critiques to help improve the clarity and quality of the manuscript:

MAJOR COMMENTS:

(1) In their analyses, Church et al. control for body size by calculating phylogenetically controlled residuals of egg size versus body size and residuals of ovariole number versus body size. They then use these residuals to assess the relationship between ovariole number versus body-size. The reason why body-size is frequently “controlled for” is to ensure that the correlation between two variables of interest (in this case ovariole number and egg size) is not due to a third variable (in this case body size). However, at the beginning of the manuscript, the reader is left to assume that there exists a significant relationship between body size and ovariole number and body size and egg size, which must be accounted for. But on lines 222 to 227 (and figure S5), the reader surprisingly learns that there is in fact no significant relationship between ovariole number and body size (with the exception of orthopterans). This leaves that reader wondering about whether there is in fact a significant relationship between egg size and body size, which is actually never shown or reported. Therefore, it becomes unclear whether it is actually necessary to control for body size and using residuals in the first place.

Furthermore, demonstrating whether there is (or is not) a relationship between egg size and body size is a key consideration because from our humble understanding of the trade-off between offspring number and offspring size, is that offspring size is assumed to be correlated to eventual adult size. So, the tradeoff as we understand it would be as follows: the more offspring,

the less energy available to invest in the offspring and so the offspring develop into smaller adults (and vice versa). However, if there is no relationship between egg size and adult size, then egg size is not a good proxy of eventual adult size, and is therefore an indicator of another ecological trait that is independent of size in general. If there is a significant indicator of eventual adult size, then we find it confusing why we would 'control' for body size in this case when it is the very thing the tradeoff is trying to assess. Perhaps we have missed something in our logic, so it would be very helpful if this was made more clear in the manuscript and laid out more clearly both in the introduction and the methods.

(2) In some panels, the Authors represent species-level regressions, while in others they present genus-level regressions, and in the main text (lines 104 to 107) they refer to family-level regressions. There are two points for consideration: the first is that the figures should be better labeled as to not confuse readers at which level the regressions are being performed, and the second (more important) consideration is that in the phylogenetic comparative method literature, there is an important pattern they call the "taxon-level affect", which is discussed extensively in Harvey and Pagel (1991) and in the brain to body scaling literature. Therefore, the authors should be aware of the taxon-level effect, discuss it, and perhaps try to explain why at the genus level there appears to be a relationship but that the species-level there is not. This result was dismissed by the Authors as not significant, but it may be more telling than the Authors may realize.

(3) In the section on "Modeling ovariole number evolution" the Authors mainly evaluate a Brownian Motion model and conclude that "a multi-rate Brownian motion model far outperforms a single rate model in fitting the data..." While this is entirely valid, we feel that the analyses and results would be more compelling if the authors considered other models of evolutionary change to explain the existence of variable and invariant insect lineages.

(4) The title of the manuscript states that the Authors' study "highlights the role of developmental constraint in life-history evolution". However there is no mention of the term 'developmental constraint' within the manuscript. It would benefit the discussion to define the concept of 'developmental constraint' and contextualise its place in the literature on evolutionary theory. Particularly, the Authors' finding that low-variation lineages tend to have little variation in ovariole number not just between species, but also within species and between the two ovaries of an individual, is very interesting and warrants further discussion, perhaps in the context of an expanded discussion of developmental constraint. For instance, do the authors view a lack of variation in intraspecific and within-individual ovariole number as evidence of more robust/less plastic ovary development in these species (perhaps due to developmental mechanisms different from the *Drosophila* paradigm, as they discuss)? How do the authors propose that differences in ovary development might drive or bias evolution of ovariole number across species?

MINOR COMMENTS:

Introduction:

(1) On lines 50 to 51, the Authors state that there is both intra- and interspecific variation of ovariole number across species. However, one of the main assumptions for phylogenetic comparative methods (which don't explicitly take into account intraspecific variation) is that interspecific variation is significantly greater than intraspecific variation. Therefore, the Authors should make this very clear.

(2) The Introduction is quite short. Perhaps Authors would like to mention / discuss whether there are other studies that have used developmental traits to assess the trade off between offspring number and size, and mention what the consensus is based on previous studies. Is this the first test? It would be useful for the reader to situate the results of this study in the context of previous work.

Methods:

- (1) Lines 73 to 74 is unclear, perhaps rearrange order of sentence.
- (2) In line 148, 'both' seems to be a typo.
- (3) It is not clear how the Authors dealt with data on intra-specific ovariole number variation. In the methods there is no explicit mention of generating measures of intra-specific variation, nor is such data presented in any of the figures, but in lines 329-331 of the results the Authors state that "invariant lineages have near-zero variation when comparing between species, between individuals within a species, and between the left and right ovary within an individual." The Authors should point to the data that support their claim of lower within-species variation in these lineages.

Results:

- (1) Lines 224 to 225 the figure and table number are missing.
- (2) The Authors should ensure the link to their dataset is correct (doi:10.5061/dryad.59zw3r253).

Referee: 3

Comments to the Author(s)

The main aim of "Repeated loss of variation in insect ovary morphology highlights the role of developmental constraint in life-history evolution" by Church et al. is to test a well-known life history prediction, namely that there is a trade off between the number and size of offspring using developmental traits (ovariole number and egg size). The authors use an impressive comparative database that they have compiled from the literature and use sophisticated phylogenetic comparative methods to test this hypothesis. They conclude that the hypothesized relationship is not general across insects, but rather a characteristic of particular groups, like the Drosophilidae. They also explore the relationship between nurse cell arrangement and egg size, which was also insignificant, as well as the dynamics of ovariole number evolution across insects. The most interesting result is that ovariole number is significantly variable in some groups and invariant in others, and that the invariant groups evolved several times independently. This last result, in our view, will be of interest to the broad readership of PRS-B and we recommend its publication. However, we would like the Authors to address the following comments and critiques to help improve the clarity and quality of the manuscript:

MAJOR COMMENTS:

- (1) In their analyses, Church et al. control for body size by calculating phylogenetically controlled residuals of egg size versus body size and residuals of ovariole number versus body size. They then use these residuals to assess the relationship between ovariole number versus body-size. The reason why body-size is frequently "controlled for" is to ensure that the correlation between two variables of interest (in this case ovariole number and egg size) is not due to a third variable (in this case body size). However, at the beginning of the manuscript, the reader is left to assume that there exists a significant relationship between body size and ovariole number and body size and egg size, which must be accounted for. But on lines 222 to 227 (and figure S5), the reader surprisingly learns that there is in fact no significant relationship between ovariole number and body size (with the exception of orthopterans). This leaves that reader wondering about whether there is in fact a significant relationship between egg size and body size, which is actually never shown or reported. Therefore, it becomes unclear whether it is actually necessary to control for body size and using residuals in the first place.

Furthermore, demonstrating whether there is (or is not) a relationship between egg size and body size is a key consideration because from our humble understanding of the trade-off between offspring number and offspring size, is that offspring size is assumed to be correlated to eventual

adult size. So, the tradeoff as we understand it would be as follows: the more offspring, the less energy available to invest in the offspring and so the offspring develop into smaller adults (and vice versa). However, if there is no relationship between egg size and adult size, then egg size is not a good proxy of eventual adult size, and is therefore an indicator of another ecological trait that is independent of size in general. If there is a significant indicator of eventual adult size, then we find it confusing why we would 'control' for body size in this case when it is the very thing the tradeoff is trying to assess. Perhaps we have missed something in our logic, so it would be very helpful if this was made more clear in the manuscript and laid out more clearly both in the introduction and the methods.

(2) In some panels, the Authors represent species-level regressions, while in others they present genus-level regressions, and in the main text (lines 104 to 107) they refer to family-level regressions. There are two points for consideration: the first is that the figures should be better labeled as to not confuse readers at which level the regressions are being performed, and the second (more important) consideration is that in the phylogenetic comparative method literature, there is an important pattern they call the "taxon-level affect", which is discussed extensively in Harvey and Pagel (1991) and in the brain to body scaling literature. Therefore, the authors should be aware of the taxon-level effect, discuss it, and perhaps try to explain why at the genus level there appears to be a relationship but that the species-level there is not. This result was dismissed by the Authors as not significant, but it may be more telling than the Authors may realize.

(3) In the section on "Modeling ovariole number evolution" the Authors mainly evaluate a Brownian Motion model and conclude that "a multi-rate Brownian motion model far outperforms a single rate model in fitting the data..." While this is entirely valid, we feel that the analyses and results would be more compelling if the authors considered other models of evolutionary change to explain the existence of variable and invariant insect lineages.

(4) The title of the manuscript states that the Authors' study "highlights the role of developmental constraint in life-history evolution". However there is no mention of the term 'developmental constraint' within the manuscript. It would benefit the discussion to define the concept of 'developmental constraint' and contextualise its place in the literature on evolutionary theory. Particularly, the Authors' finding that low-variation lineages tend to have little variation in ovariole number not just between species, but also within species and between the two ovaries of an individual, is very interesting and warrants further discussion, perhaps in the context of an expanded discussion of developmental constraint. For instance, do the authors view a lack of variation in intraspecific and within-individual ovariole number as evidence of more robust/less plastic ovary development in these species (perhaps due to developmental mechanisms different from the *Drosophila* paradigm, as they discuss)? How do the authors propose that differences in ovary development might drive or bias evolution of ovariole number across species?

MINOR COMMENTS:

Introduction:

(1) On lines 50 to 51, the Authors state that there is both intra- and interspecific variation of ovariole number across species. However, one of the main assumptions for phylogenetic comparative methods (which don't explicitly take into account intraspecific variation) is that interspecific variation is significantly greater than intraspecific variation. Therefore, the Authors should make this very clear.

(2) The Introduction is quite short. Perhaps Authors would like to mention / discuss whether there are other studies that have used developmental traits to assess the trade off between offspring number and size, and mention what the consensus is based on previous studies. Is this the first test? It would be useful for the reader to situate the results of this study in the context of previous work.

Methods:

- (1) Lines 73 to 74 is unclear, perhaps rearrange order of sentence.
- (2) In line 148, 'both' seems to be a typo.
- (3) It is not clear how the Authors dealt with data on intra-specific ovariole number variation. In the methods there is no explicit mention of generating measures of intra-specific variation, nor is such data presented in any of the figures, but in lines 329-331 of the results the Authors state that "invariant lineages have near-zero variation when comparing between species, between individuals within a species, and between the left and right ovary within an individual." The Authors should point to the data that support their claim of lower within-species variation in these lineages.

Results:

- (1) Lines 224 to 225 the figure and table number are missing.
- (2) The Authors should ensure the link to their dataset is correct (doi:10.5061/dryad.59zw3r253).

Author's Response to Decision Letter for (RSPB-2021-0150.R0)

See Appendix A.

Decision letter (RSPB-2021-0150.R1)

26-Mar-2021

Dear Dr Church

I am pleased to inform you that your Review manuscript RSPB-2021-0150.R1 entitled "Repeated loss of variation in insect ovary morphology highlights the role of development in life-history evolution" has been accepted for publication in Proceedings B.

The referee(s) do not recommend any further changes. Therefore, please proof-read your manuscript carefully and upload your final files for publication. Because the schedule for publication is very tight, it is a condition of publication that you submit the revised version of your manuscript within 7 days. If you do not think you will be able to meet this date please let me know immediately.

To upload your manuscript, log into <http://mc.manuscriptcentral.com/prsb> and enter your Author Centre, where you will find your manuscript title listed under "Manuscripts with Decisions." Under "Actions," click on "Create a Revision." Your manuscript number has been appended to denote a revision.

You will be unable to make your revisions on the originally submitted version of the manuscript. Instead, upload a new version through your Author Centre.

1) A text file of the manuscript (doc, txt, rtf or tex), including the references, tables (including captions) and figure captions. Please remove any tracked changes from the text before submission. PDF files are not an accepted format for the "Main Document".

2) A separate electronic file of each figure (tiff, EPS or print-quality PDF preferred). The format should be produced directly from original creation package, or original software format. Please note that PowerPoint files are not accepted.

3) Electronic supplementary material: this should be contained in a separate file from the main text and the file name should contain the author's name and journal name, e.g. authorname_procb_ESM_figures.pdf

All supplementary materials accompanying an accepted article will be treated as in their final form. They will be published alongside the paper on the journal website and posted on the online figshare repository. Files on figshare will be made available approximately one week before the accompanying article so that the supplementary material can be attributed a unique DOI. Please see: <https://royalsociety.org/journals/authors/author-guidelines/>

4) Data-Sharing and data citation

It is a condition of publication that data supporting your paper are made available. Data should be made available either in the electronic supplementary material or through an appropriate repository. Details of how to access data should be included in your paper. Please see <https://royalsociety.org/journals/ethics-policies/data-sharing-mining/> for more details.

If you wish to submit your data to Dryad (<http://datadryad.org/>) and have not already done so you can submit your data via this link <http://datadryad.org/submit?journalID=RSPB&manu=RSPB-2021-0150.R1> which will take you to your unique entry in the Dryad repository.

Once again, thank you for submitting your manuscript to Proceedings B and I look forward to receiving your final version. If you have any questions at all, please do not hesitate to get in touch.

Sincerely,
Professor Gary Carvalho
Editor, Proceedings B
<mailto:proceedingsb@royalsociety.org>

Associate Editor Board Member

Comments to Author:

I enjoyed reading this manuscript again and commend the authors on their efforts to respond to the reviewers' constructive suggestions and incorporate these, where feasible, into their report. Congratulations - I am confident that this will be of great interest to readers of Proceedings B.

I only have some very minor final comments on phrasing or to slightly improve clarity. I note also that Proceedings B require any data/code to be available during peer review (<https://royalsociety.org/journals/authors/author-guidelines/#data>), and Dryad allow for such sharing without public release of the dataset (<https://datadryad.org/stash/faq#ppr>). I encourage the authors to do this for future submissions.

Minor suggestions:

- Add short phrase to explain 'nurse cells' in Abstract; for the broad readership of Proceedings B.
- 1.76 "The taxonomic groups used to search" -> "The taxonomic groups used in the search process [or exercise]"
- 1.79 Should 'ten publications' be in parentheses? The sentence does not make sense otherwise.
- It would be helpful in the methods or results to give summary of the dataset: ovariole number was established for how many unique species, how many genera across how many families? (this could help put the values of 3355 records across 448 publications into more relevant context, this could also be mentioned at 1.376)
- 1.214 would add 'across genera' after 'negative relationship between egg size and ovariole number' (Note that Reviewer 2's major comment 2 had been more clarity on the level of analysis in the figure legends, this should also be reflected in the main text)
- 1.285 delete comma after 'nurse cells'

Author's Response to Decision Letter for (RSPB-2021-0150.R1)

See Appendix B.

Decision letter (RSPB-2021-0150.R2)

06-Apr-2021

Dear Dr Church

I am pleased to inform you that your manuscript entitled "Repeated loss of variation in insect ovary morphology highlights the role of development in life-history evolution" has been accepted for publication in Proceedings B.

Data Accessibility section

Open Access

You are invited to opt for Open Access, making your freely available to all as soon as it is ready for publication under a CC BY licence. Our article processing charge for Open Access is £1700. Corresponding authors from member institutions

Paper charges

Sincerely,
Editor, Proceedings B
<mailto:proceedingsb@royalsociety.org>

Appendix A

[revised manuscript text omitted]

**contrast**, our previous understanding of how ovariole number is regulated comes from
research on *Drosophila melanogaster*, where the number of ovarioles can vary between the
left and right ovaries within an individual, as well as across individuals within a
population^{57,58}. In this species, adult ovariole number is determined by cell proliferation
and rearrangement during larval development^{59,60}. Variation in adult number is derived
primarily from variation in the number of “terminal filament precursor cells”^{61,62}, as well as
from variation in the number of those precursor cells that group together to form the
structure that initiates ovariole formation, known as a “terminal filament”⁶³. Across species
of *Drosophila*, variation in average adult ovariole number results primarily from variation
in the average number of terminal filament precursor cells⁶².
When considering the developmental processes that could give rise to invariant ovariole
number, we propose that the major determinants of ovariole number known from
*Drosophila* may not apply. To achieve an invariant ovariole number, these processes might
instead include mechanisms for strict counting of individual cells or discrete cell
subpopulations. In the former, if the cells that ultimately comprised a terminal filament
were derived by mitotic division from a single progenitor, rather than by cellular
rearrangements as is the case in *Drosophila*⁵⁹, then an invariant ovariole number could be
achieved via strict control of the number of precursor cells. Alternatively, an invariant
ovariole number could be achieved by partitioning the starting population of precursor
cells into a tightly regulated number of subpopulations. This would again be a departure
from known mechanisms in *Drosophila*, in which a variable number of precursor cells are
gathered into terminal filaments until the population is depleted^{59,63}. The determining
factor for partitioning the precursor pool could be, for example, a spatially variable
morphogen emanating from adjacent tissues⁶⁴ or a reaction-diffusion patterning process⁶⁵
within the developing ovary, as these have been shown to generate fixed numbers of
multicellular structures in other developmental contexts⁶⁶⁻⁶⁸. These predictions could be
tested by characterizing the dynamics of cell number and position across invariant
lineages, and making comparisons to corresponding data from their variable relatives.
The evolutionary transitions between variable and invariant ovariole number are
reminiscent of other quantitative traits across multicellular life, including patterns of
variability and invariance in arthropod segment number^{69,70}, vertebrate digit number^{71,72},
or the number of angiosperm floral organs^{73,74}. Across these systems, the evolutionary
history of morphogenetic counting mechanisms is poorly understood. We suggest that
insect ovariole number presents an ideal case to study this phenomenon. In particular, we
note the evidence that invariance has evolved convergently at least nine times, as well as
the evidence of several reversals back to variability from an invariant ancestral state (Fig.
5). These convergent lineages provide an opportunity to test the predictability of
evolutionary changes to counting mechanisms, by asking whether convergent evolution of
invariance involves convergent canalization of shared molecular mechanisms.
**Acknowledgements**
This work was supported by NSF GRFP DGE1745303 to SHC and funds from Harvard
University to support SHC and CGE. NM was supported by the E3 REU program at Harvard
University. We thank members of the Extavour Lab for discussion.
**Author Contributions**
SHC led the data collection, analysis, and writing of the manuscript. NM performed the
initial literature search, data collection, and analysis. BASdM assembled all phylogenies for
analysis. SD, BASdM, and CGE contributed to data analysis, visualization, discussion, and
writing.
**Competing Interests**
The authors declare no competing interests.
**References**
- 1. Stearns, S. C. *The evolution of life histories*. (Oxford University Press, 1992).
- 2. Smith, C. C. & Fretwell, S. D. The optimal balance between size and number of offspring.
*The American Naturalist* **108**, 499–506 (1974).
- 3. Lack, D. The significance of clutch-size. *Ibis* **89**, 302–352 (1947).
- 4. Honěk, A. Intraspecific variation in body size and fecundity in insects: A general
relationship. *Oikos* **66**, 483–492 (1993).
- 5. Berrigan, D. The allometry of egg size and number in insects. *Oikos* **60**, 313–321 (1991).
- 6. Hodin, J. She shapes events as they come: Plasticity in female insect reproduction. in
*Phenotypic plasticity of insects : Mechanisms and consequences* (eds. Whitman, Douglas W &
Ananthakrishnan, T. N.) 423–521 (Science Publishers, 2009).
- 7. Büning, J. *The insect ovary: Ultrastructure, previtellogenic growth and evolution*. (Springer
Science & Business Media, 1994).
- 8. Iwata, K. The comparative anatomy of the ovary in Hymenoptera. Part I. Aculeata. *Mushi*
**29**, 1–37 (1955).
- 9. Church, S. H., Donoughe, S., Medeiros, B. A. de & Extavour, C. G. A dataset of egg size and
shape from more than 6,700 insect species. *Scientific Data* **6**, 1–11 (2019).
- 10. Gilbert, J. D. J. *PhD Thesis: The evolution of parental care in insects*. (University of
Cambridge, 2007).
- 11. Gilbert, J. D. & Manica, A. Parental care trade-offs and life-history relationships in
insects. *The American Naturalist* **176**, 212–226 (2010).
- 12. Rainford, J. L., Hofreiter, M. & Mayhew, P. J. Phylogenetic analyses suggest that
diversification and body size evolution are independent in insects. *BMC Evolutionary*
*Biology* **16**, 8 (2016).
- 13. Church, S. H., Donoughe, S., Medeiros, B. A. de & Extavour, C. G. Insect egg size and shape
evolve with ecology but not developmental rate. *Nature* **571**, 58–62 (2019).
- 14. Medeiros, B. A. S. de. TaxReformer. <https://github.com/brunoasm/TaxReformer>
(2019).
- 15. Su, X. H. *et al.* Testicular development and modes of apoptosis during spermatogenesis
in various castes of the termite *Reticulitermes labralis* (Isoptera: Rhinotermitidae).
*Arthropod Structure and Development* **44**, 630–638 (2015).
- 16. Hernandez, L. C., Fajardo, G., Fuentes, L. S. & Comoglio, L. Biology and reproductive
traits of *Drymoea veliterna* (druce, 1885) (Lepidoptera: Geometridae). *Journal of Insect*
*Biodiversity* **5**, 1–9 (2017).
- 17. Gilbert, J. D. J. Insect dry weight: Shortcut to a difficult quantity using museum
specimens. *Florida Entomologist* **94**, 964–970 (2011).
- 18. Iwata, K. Large-sized eggs in Curculionoidea (Coleoptera). *Research Bulletin of Hyogo*
*Agricultural College* **7**, 43–45 (1966).
- 19. Iwata, K. & Sakagami, S. F. Gigantism and dwarfism in bee eggs in relation to the mode
of life, with notes on the number of ovarioles. *Japanese Journal of Ecology* **16**, 4–16 (1966).
- 20. Waloff, N. Number and development of ovarioles of some Acridoidea (Orthoptera) in
relation to climate. *Physiologia Comparata et Oecologia* vol. 3 370–390 (1954).
- 21. Starmer, W. T. *et al.* Phylogenetic, geographical, and temporal analysis of female
reproductive trade-offs in Drosophilidae. *Evolutionary Biology* **33**, 139–171 (2003).
- 22. Reinhardt, K., Köhler, G., Maas, S. & Detzel, P. Low dispersal ability and habitat
specificity promote extinctions in rare but not in widespread species: The Orthoptera of
Germany. *Ecography* **28**, 593–602 (2005).
- 23. Rainford, J. L., Hofreiter, M., Nicholson, D. B. & Mayhew, P. J. Phylogenetic distribution of
extant richness suggests metamorphosis is a key innovation driving diversification in
insects. *PLoS One* **9**, e109085 (2014).
- 24. Paradis, E., Claude, J. & Strimmer, K. APE: Analyses of phylogenetics and evolution in R
language. *Bioinformatics* **20**, 289–290 (2004).
- 25. Pinheiro, J., Bates, D., DebRoy, S. & Sarkar, D. R Core Team (2014) nlme: Linear and
nonlinear mixed effects models. R package version 3.1-117. Available at [http://cran.r-](http://cran.r-project.org/package=nlme)
[project.org/package=nlme](http://cran.r-project.org/package=nlme) (2014).
- 26. Revell, L. J. Size-correction and principal components for interspecific comparative
studies. *Evolution: International Journal of Organic Evolution* **63**, 3258–3268 (2009).
- 27. Harmon, L. J., Weir, J. T., Brock, C. D., Glor, R. E. & Challenger, W. GEIGER: Investigating
evolutionary radiations. *Bioinformatics* **24**, 129–131 (2007).
- 28. Tung Ho, L. S. & Ané, C. A linear-time algorithm for Gaussian and non-gaussian trait
evolution models. *Systematic Biology* **63**, 397–408 (2014).
- 29. Beaulieu, J. M., O'Meara, B. C. & Donoghue, M. J. Identifying hidden rate changes in the
evolution of a binary morphological character: The evolution of plant habit in campanulid
angiosperms. *Systematic Biology* **62**, 725–737 (2013).
- 30. Beaulieu, J. M., Jhwueng, D.-C., Boettiger, C. & O'Meara, B. C. Modeling stabilizing
selection: Expanding the Ornstein–Uhlenbeck model of adaptive evolution. *Evolution* **66**,
2369–2383 (2012).
- 31. Pennell, M. W., FitzJohn, R. G., Cornwell, W. K. & Harmon, L. J. Model adequacy and the
macroevolution of angiosperm functional traits. *The American Naturalist* **186**, E33–E50
(2015).
- 32. Rabosky, D. L. Automatic detection of key innovations, rate shifts, and diversity-
dependence on phylogenetic trees. *PLoS One* **9**, e89543 (2014).
- 33. Rabosky, D. L. *et al.* BAMM tools: An R package for the analysis of evolutionary
dynamics on phylogenetic trees. *Methods In Ecology and Evolution* **5**, 701–707 (2014).
- 34. Bugnion, É. & Popoff, N. Anatomie de la reine et du roi-termite. *Mémoires De La société*
*Zoologique De France* **25**, 210–232 (1912).
- 35. Robertson, H. Sperm transfer in the ant *Carebara vidua* F. Smith (Hymenoptera:
Formicidae). *Insectes Sociaux* **42**, 411–418 (1995).
- 36. Schneirla, T. A comparison of species and genera in the ant subfamily Dorylinae with
respect to functional pattern. *Insectes Sociaux* **4**, 259–298 (1957).
- 37. Büning, J. The trophic tissue of telotrophic ovarioles in polyphage Coleoptera.
*Zoomorphologie* **93**, 33–50 (1979).
- 38. Richter, P. & Baker, C. Ovariole number in Scarabaeoidea (Coleoptera: Lucanidae,
Passalidae, Scarabaeidae). *Proceedings of The Entomological Society of Washington* **76**, 480–
498 (1974).
- 39. Meier, R., Kotrba, M. & Ferrar, P. Ovoviviparity and viviparity in the Diptera. *Biological*
*Reviews* **74**, 199–258 (1999).
- 40. Pollock, J. Viviparous adaptations in the acalyptrate genera *Pachylophus* (Chloropidae)
and *Cyrtona* (Curtonotidae) (Diptera: Schizophora). *Annals of The Natal Museum* **37**, 183–
189 (1996).
- 41. Pluot, D. Évolution régressive des ovarioles chez les coléoptères Scarabaeinae. *Annales*
*de la Société Entomologique de France* **15**, 575–588 (1979).
- 42. Montague, J. R., Mangan, R. L. & Starmer, W. T. Reproductive allocation in the Hawaiian
Drosophilidae: Egg size and number. *The American Naturalist* **118**, 865–871 (1981).
- 43. Stewart, L., Hemptinne, J.-L. & Dixon, A. Reproductive tactics of ladybird beetles:
Relationships between egg size, ovariole number and developmental time. *Functional*
*Ecology* **5**, 380–385 (1991).
- 44. David, J. Nombre d'ovarioles chez *Drosophila melanogaster*: Relation avec la fécondité et
valeur adaptative. *Archives De Zoologie Expérimentale et Générale* (1970).
- 45. Boulétreau-Merle, J., Allemand, R., Cohet, Y. & David, J. Reproductive strategy in
*Drosophila melanogaster*: Significance of a genetic divergence between temperate and
tropical populations. *Oecologia* **53**, 323–329 (1982).
- 46. Schmidt, P. S., Matzkin, L., Ippolito, M. & Eanes, W. F. Geographic variation in diapause
incidence, life-history traits, and climatic adaptation in *Drosophila melanogaster*. *Evolution*
**59**, 1721–1732 (2005).
- 47. Wayne, M. L., Hackett, J. B. & Mackay, T. F. Quantitative genetics of ovariole number in
*Drosophila melanogaster*. I. Segregating variation and fitness. *Evolution* **51**, 1156–1163
(1997).
- 48. King, R. C. *Ovarian development in Drosophila melanogaster*. (Academic Press, 1970).
- 49. Blomberg, S. P., Rathnayake, S. I. & Moreau, C. M. Beyond brownian motion and the
ornstein-uhlenbeck process: Stochastic diffusion models for the evolution of quantitative
characters. *The American Naturalist* **195**, 145–165 (2020).
- 50. Koch, L. K. & Meunier, J. Mother and offspring fitness in an insect with maternal care:
Phenotypic trade-offs between egg number, egg mass and egg care. *BMC Evolutionary*
*Biology* **14**, 125 (2014).
- 51. Drummond-Barbosa, D. & Spradling, A. C. Stem cells and their progeny respond to
nutritional changes during *Drosophila* oogenesis. *Developmental Biology* **231**, 265–278
(2001).
- 52. Ables, E. T., Laws, K. M. & Drummond-Barbosa, D. Control of adult stem cells in vivo by a
dynamic physiological environment: Diet-dependent systemic factors in *Drosophila* and
beyond. *Wiley Interdisciplinary Reviews: Developmental Biology* **1**, 657–674 (2012).
- 53. Mirth, C. K., Alves, A. N. & Piper, M. D. Turning food into eggs: Insights from nutritional
biology and developmental physiology of *Drosophila*. *Current opinion in insect science* **31**,
49–57 (2019).
- 54. Wcislo, W. T. The roles of seasonality, host synchrony, and behaviour in the evolutions
and distributions of nest parasites in Hymenoptera (Insecta), with special reference to bees
(Apoidea). *Biological Reviews* **62**, 515–543 (1987).
- 55. Partridge, L., Fowler, K., Trevitt, S. & Sharp, W. An examination of the effects of males on
the survival and egg-production rates of female *Drosophila melanogaster*. *Journal of Insect*
*Physiology* **32**, 925–929 (1986).
- 56. Parker, G. & Courtney, S. Models of clutch size in insect oviposition. *Theoretical*
*Population Biology* **26**, 27–48 (1984).
- 57. Telonis-Scott, M., McIntyre, L. & Wayne, M. Genetic architecture of two fitness-related
traits in *Drosophila melanogaster*: Ovariole number and thorax length. *Genetica* **125**, 211–
222 (2005).
- 58. Bergland, A. O., Genissel, A., Nuzhdin, S. V. & Tatar, M. Quantitative trait loci affecting
phenotypic plasticity and the allometric relationship of ovariole number and thorax length
in *Drosophila melanogaster*. *Genetics* **180**, 567–582 (2008).
- 59. Godt, D. & Laski, F. A. Mechanisms of cell rearrangement and cell recruitment in
*Drosophila* ovary morphogenesis and the requirement of *bric a brac*. *Development* **121**,
173–187 (1995).
- 60. King, R. C., Aggarwal, S. K. & Aggarwal, U. The development of the female *Drosophila*
reproductive system. *Journal of Morphology* **124**, 143–165 (1968).
- 61. Green II, D. A. & Extavour, C. G. Convergent evolution of a reproductive trait through
distinct developmental mechanisms in *Drosophila*. *Developmental Biology* **372**, 120–130
(2012).
- 62. Sarikaya, D. P. *et al.* Reproductive capacity evolves in response to ecology through
common changes in cell number in Hawaiian *Drosophila*. *Current Biology* **29**, 1877–1884
(2019).
- 63. Sarikaya, D. P. *et al.* The roles of cell size and cell number in determining ovariole
number in *Drosophila*. *Developmental Biology* **363**, 279–289 (2012).
- 64. Lawrence, P. A. & Struhl, G. Morphogens, compartments, and pattern: Lessons from
*Drosophila*? *Cell* **85**, 951–961 (1996).
- 65. Kondo, S. & Miura, T. Reaction-diffusion model as a framework for understanding
biological pattern formation. *science* **329**, 1616–1620 (2010).
- 66. Salazar-Ciudad, I. Tooth morphogenesis in vivo, in vitro, and in silico. *Current Topics in*
*Developmental Biology* **81**, 341–371 (2008).
- 67. Hatini, V. & DiNardo, S. Divide and conquer: Pattern formation in drosophila embryonic
epidermis. *Trends in Genetics* **17**, 574–579 (2001).
- 68. Clark, E., Peel, A. D. & Akam, M. Arthropod segmentation. *Development* **146**, dev170480
(2019).
- 69. Arthur, W. & Farrow, M. The pattern of variation in centipede segment number as an
example of developmental constraint in evolution. *Journal of Theoretical Biology* **200**, 183–
191 (1999).
- 70. Vedel, V., Chipman, A. D., Akam, M. & Arthur, W. Temperature-dependent plasticity of
segment number in an arthropod species: The centipede *Strigamia maritima*. *Evolution &*
*Development* **10**, 487–492 (2008).
- 71. Holder, N. Developmental constraints and the evolution of vertebrate digit patterns.
*Journal of Theoretical Biology* **104**, 451–471 (1983).
- 72. Saxena, A., Towers, M. & Cooper, K. L. The origins, scaling and loss of tetrapod digits.
*Philosophical Transactions of the Royal Society B: Biological Sciences* **372**, 20150482 (2017).
- 73. Ambrose, B. A. & Purugganan, M. *The evolution of plant form*. vol. 45 (Wiley, 2012).
- 74. Kitazawa, M. S. & Fujimoto, K. A developmental basis for stochasticity in floral organ
numbers. *Frontiers in Plant Science* **5**, 545 (2014).

Appendix B

April 2, 2021

Dear Dr. Carvalho,

Thank you for your attention to the details included in this request for minor revisions. We have addressed all of the requested changes, and our responses are included below.

Sincerely,
Samuel H. Church

Minor suggestions:

- Add short phrase to explain 'nurse cells' in Abstract; for the broad readership of Proceedings B.

RESPONSE: We have added the following phrase: "cells that contribute nutrients and patterning information during oogenesis (nurse cells)" l.24

- l.76 "The taxonomic groups used to search" -> "The taxonomic groups used in the search process [or exercise]"

RESPONSE: We have made the requested change

- l.79 Should 'ten publications' be in parentheses? The sentence does not make sense otherwise.

RESPONSE: We have corrected this error. The sentence now reads "we evaluated the first ten publications in the search results"

- It would be helpful in the methods or results to give summary of the dataset: ovariole number was established for how many unique species, how many genera across how many families? (this could help put the values of 3355 records across 448 publications into more relevant context, this could also be mentioned at l.376)

RESPONSE: We have included the number of orders, families, and species in the introduction (l.55), methods (l.83), and discussion (only species number, l.378).

- l.214 would add 'across genera' after 'negative relationship between egg size and ovariole number' (Note that Reviewer 2's major comment 2 had been more clarity on the level of analysis in the figure legends, this should also be reflected in the main text)

RESPONSE: We have made the requested change.

- l.285 delete comma after 'nurse cells'

RESPONSE: We have corrected this error.